# Persisting effects of jaw clenching on dynamic steady-state balance

Cagla Fadillioglu[1]*, Lisa Kanus[2], Felix Möhler[1], Steffen Ringhof[3,4], Marc Schmitter[2], Daniel Hellmann[2,5], Thorsten Stein[1]

1 BioMotion Center, Institute of Sports and Sports Science, Karlsruhe Institute of Technology (KIT), Karlsruhe, Germany, 2 Department of Prosthodontics, University of Würzburg, Würzburg, Germany, 3 Department of Sport and Sport Science, University of Freiburg, Freiburg, Germany, 4 Department of Diagnostic and Interventional Radiology, University Medical Center Freiburg, Faculty of Medicine, University of Freiburg, Freiburg, Germany, 5 Dental Academy for Continuing Professional Development, Karlsruhe, Germany

◉ These authors contributed equally to this work.
* cagla.fadillioglu@kit.edu

**Data Availability Statement:** The data required to replicate all study findings reported in the article can be found under the Supporting Information.

**Funding:** This work was supported by the German Research Foundation [STE 2093/4-3 & SCHM 2456/6-3]. The funders had no role in study

## Abstract

The effects of jaw clenching on balance has been shown under static steady-state conditions but the effects on dynamic steady-state balance have not yet been investigated. On this basis, the research questions were: 1) if jaw clenching improves dynamic steady-state balance; 2) if the effects persist when the jaw clenching task loses its novelty and the increased attention associated with it; 3) if the improved dynamic steady-state balance performance is associated with decreased muscle activity. A total of 48 physically active healthy adults were assigned to three groups differing in intervention (Jaw clenching and balance training (JBT), only balance training (OBT) or the no-training control group (CON)) and attending two measurement points separated by two weeks. A stabilometer was used to assess the dynamic steady-state balance performance in a jaw clenching and non-clenching condition. Dynamic steady-state balance performance was measured by the time at equilibrium (TAE). The activities of tibialis anterior (TA), gastrocnemius medialis (GM), rectus femoris (RF), biceps femoris (BF) and masseter (MA) muscles were recorded by a wireless EMG system. Integrated EMG (iEMG) was calculated to quantify the muscle activities. All groups had better dynamic steady-state balance performance in the jaw clenching condition than non-clenching at T1, and the positive effects persisted at T2 even though the jaw clenching task lost its novelty and attention associated with it after balance training with simultaneous jaw clenching. Independent of the intervention, all groups had better dynamic steady-state balance performances at T2. Moreover, reductions in muscle activities were observed at T2 parallel to the dynamic steady-state balance performance improvement. Previous studies showed that jaw clenching alters balance during upright standing, predictable perturbations when standing on the ground and unpredictable perturbations when standing on an oscillating platform. This study complemented the previous findings by showing positive effects of jaw clenching on dynamic steady-state balance performance.

design, data collection and analysis, decision to publish, or preparation of the manuscript.

**Competing interests:** The authors have declared that no competing interests exist.

## Introduction

The postural control system regulates the body's position with respect to the environment for the dual purposes of balance and orientation [1]. Good balance is crucial for daily activities and is associated with decreased risk of falls [2] and injuries [3]. Therefore, the methods to improve postural control, such as balance training [4], are highly appreciated. However, balance is not a general ability but task-specific [5]. Balance can generally be classified as static steady-state, dynamic steady-state, dynamic reactive and dynamic proactive based on the performed activity [6]. Static steady-state balance basically comprises unperturbed conditions, such as during quiet upright standing, whereas dynamic steady-state balance involves the maintenance of a steady position while moving (e.g., walking). Dynamic reactive balance can be defined as the compensation of an unpredicted postural perturbation to maintain the balance. In case of proactive balance, a predicted perturbation is anticipated and compensated before balance is disturbed [7, 8]. Good balance in one of these sub-categories does not necessarily mean good balance in the others due to the task specificity of balance [7]. Against this background, the effects of balance must be investigated in individual sub-categories.

Postural control can be influenced by many factors including the status and activity of the stomatognathic system. There is a growing body of literature showing the associations between postural activities under static steady-state conditions and stomatognathic motor activities in the form of jaw clenching in different jaw relationships (e.g. maximum intercuspation or different occlusal appliances) [9–11]. Particularly regarding jaw clenching, a lower sway of body in the anterior–posterior direction [9, 11], a lower variability in muscular co-contraction patterns [10] and lower sway of trunk and head during upright standing [12] were previously reported. The effects of jaw clenching on dynamic and proactive balance [13, 14] were also shown. However, the effects of jaw clenching on dynamic steady-state balance are not well known [15].

Despite the growing evidence for a relationship between the stomatognathic system and postural activities, the underlying mechanisms are not fully understood. Several studies [e.g., 16, 17] suggested that jaw clenching may result in increased motor excitability similar to the Jendrassik maneuver [18], or an increased muscle force in association with the H-reflex mechanism [19]. Also, the co-contraction pattern of the jaw and neck muscles may help to improve postural control by contributing to a more stable head or gaze position [20]. Furthermore, neuronal links of the trigeminal nerve to the rest of the nervous system were shown in animal models [21]. Another possible explanation might be that the instruction of jaw clenching during the simultaneous performance of a balancing task might create a dual-task scenario. In this case, the attention increases due to the secondary task, and consequently automatization of postural control is enhanced [22]. Based on these findings, it may be hypothesized that simultaneous execution of the jaw clenching task improves balance performance due to its novelty and increased requirement of attention, but not specifically due to neurophysiological effects.

Previous studies showed various effects of jaw clenching during upright standing [9–12], during predictable perturbations applied when standing on the ground [14] and during unpredictable perturbations when standing on an oscillating platform [13]. However, the effects of jaw clenching during a dynamic steady-state balance task have not been fully investigated. In this study, this research gap was addressed. Using two measurement times (T1 and T2) two weeks apart, it was evaluated whether the stabilizing effects of jaw clenching persist at T2, despite the diminished novelty and competing influence of a secondary task (and therefore decreased attention). It was hypothesized that (1) jaw clenching improves dynamic steady-state balance at T1; (2) the effects persist at T2; and (3) better dynamic steady-state balance performance is associated with decreased muscle activity due to movement efficiency [23].

## Methods

### Participants

An *a priori* power analysis was conducted based on a study analyzing the effects of jaw clenching on postural stability during upright standing [12]. That analysis revealed that 16 participants per group would be enough to reach sufficient power (>0.8). On this basis, 48 healthy adults (21 female, 27 male; age: 22.9 ± 2.5 years; height: 1.74 ± 0.09 m; body mass: 70.0 ± 12.2 kg) voluntarily participated after giving written informed consent. They were physically active (active 4.2 ± 1.2 days/week and 368 ± 153 min/week), naive to the stabilometer task and had no muscular or neurological diseases. They had no signs and symptoms of temporomandibular disorders (assessed by means of the research diagnostic criteria for temporomandibular disorders [24]) and presented with full dentition (except for 3rd molars) in neutral occlusion. The recruitment period for this study was between 13.09.2021–27.07.20222. The study was approved by the Ethics Committee of the Karlsruhe Institute of Technology.

### Instrumentation

Dynamic steady-state balance was assessed using a stabilometer (Stability Platform, Model 16030, Lafayette Instrument Company, Lafayette, IN, USA) containing a 65×107 cm wooden platform with a maximum deviation of ± 15˚ (Fig 1A and 1B). EMG data of the tibialis anterior (TA), gastrocnemius medialis (GM), rectus femoris (RF), biceps femoris (BF) and masseter (MA) of the right side were recorded by a wireless EMG system (Noraxon, Scottsdale, USA; 2000 Hz). As preparation, the skin over the muscles was carefully shaved, abraded, and rinsed with alcohol. Bipolar Ag/AgCl surface electrodes (diameter 14 mm, center-to-center distance

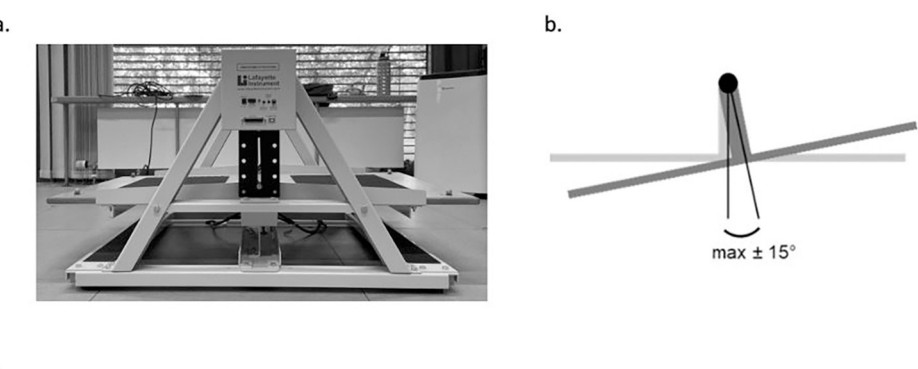

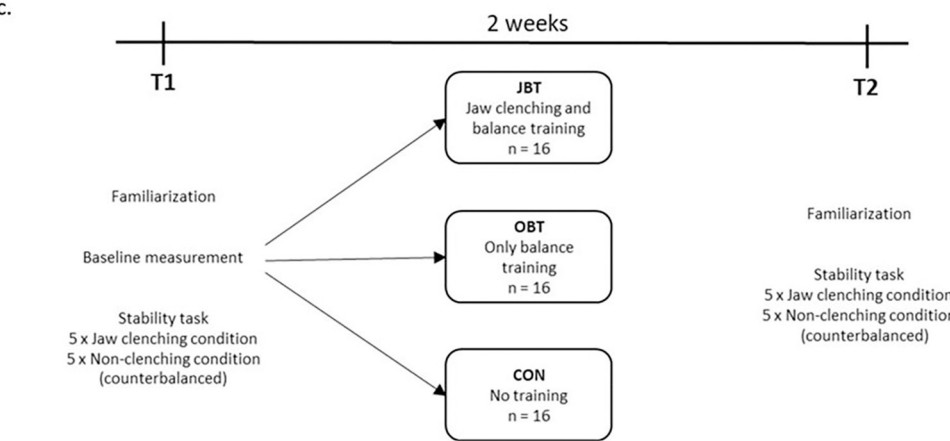

**Fig 1. a.** Stabilometer. **b.** Degrees of freedom and maximum deviation of the platform. **c.** Experimental protocol.

20 mm; Noraxon Dual Electrodes, Noraxon, Scottsdale, USA) were positioned in accordance with the European Recommendations for Surface EMG [25]. The positions of the EMG electrodes were marked with temporary tattoo ink (MyJagua, Greven, Germany) at T1 to allow identical positioning at T2.

## Protocol

The experimental protocol is illustrated in Fig 1C. First, the participants were familiarized to the stabilometer by standing on it for 1 min with rubber bands under it (the easier form of the task), then for 1 min without the rubber bands (the task to be performed during the measurements). Afterwards, a baseline measurement of 30 s was performed to determine the initial dynamic steady-state balance performance operationalized by the time at equilibrium (TAE; for details see the "Data analysis" section). Both baseline measurement result and gender were considered to assign the participants to one of three groups: jaw clenching and balance training (JBT), only balance training (OBT) or the no-training control group (CON). Statistical examination by one-way ANOVA revealed no baseline performance differences between the three groups (p = 0.982). All groups had 7 female and 9 male participants.

After warming up on a treadmill (h/p/cosmos Saturn, Nussdorf-Traunstein, Germany) for 5 min at 6 km/h, maximum voluntary contraction (MVC) tests were performed for each muscle. Just before the measurements, each participant trained with a RehaBite® (Plastyle GmbH, Uttenreuth, Germany) to become familiar with applying a submaximal force of 75 N [11]. The EMG data of MA were monitored during training to determine the corresponding muscle activity for later use as reference during the measurements [13, 15]. During the subsequent balancing task, participants clenched on an Aqualizer® intraoral splint (medium volume; Dentrade International, Cologne, Germany).

Regarding the balance task, participants were asked to keep the stabilometer platform in the horizontal position as long as possible and to focus on a target positioned at eye level and 3 m away from the center of the platform. For the jaw clenching trials, participants were asked to simultaneously clench their jaws. Five valid trials, each 30 s, were collected for each condition (clenching/non-clenching). There was a break of 30 s between each trial to avoid fatigue. The order of clenching conditions was counterbalanced within the groups and each participant was randomly assigned to an order. At T2, the same protocol as during T1 was executed except for the baseline measurement.

## Intervention

Between T1 and T2, the participants of JBT and OBT followed a two-week training program comprising six training sessions at least two days apart from each other, whereas CON did not train. Each training session was performed in the BioMotion Center under the supervision of experienced staff and lasted about 15 min. As in the measurements, participants were asked to keep the platform in the horizontal position as long as possible. In total, 10 trials (2 sets of 5 trials) of 30 s were performed in each training session. There was a break of 30 s between each trial and 2 min between each set. The participants of JBT trained in the jaw clenching condition and OBT in the non-clenching condition. In each training session, JBT additionally trained with the Rehabite® for five minutes before balance training to get used to the jaw clenching task.

## Data analysis

All data were recorded in Vicon Nexus 2.12 (Vicon Motion Systems; Oxford Metrics Group, Oxford, UK) and exported for further processing in MATLAB R2022a (MathWorks, Natick,

USA). The analog output signal of the platform was filtered with a Butterworth low-pass filter (fourth-order; cut-off frequency 10 Hz); and EMG data with a Butterworth band-pass filter (fourth-order; cut-off frequency 10–500 Hz). After filtering, EMG data were rectified and smoothed by averaging with a sliding window of 30 ms and finally normalized to the MVC references [11]. For each trial, time at equilibrium (TAE, ± 3˚ deviation from the horizontal position [23, 26] for at least 500 ms [5]) as well as time normalized iEMG for each muscle were calculated. A higher TAE was considered as better dynamic steady-state balance performance. The data required to replicate all study findings reported in the article can be found in S1 File.

## Statistics

Statistical analysis was done with IBM SPSS Statistics 29.0 (IBM Corporation, Armonk, NY, USA). Kolmogorov-Smirnov tests were performed to determine the normality of data distribution. For each measurement time and condition, the trial with the highest TAE was used for statistical tests.

For TAE at T1, a paired t-test was performed to analyze the effects of jaw clenching on dynamic steady-state balance performance (Hypothesis 1). Additionally, for each dependent parameter (i.e. TAE and iEMG), a three-factorial mixed ANOVA (3 groups x 2 clenching conditions x 2 measurement times) was conducted to test the remaining hypotheses. *Post-hoc* t-tests for pairwise group comparisons were run with Bonferroni-Holm corrections in the case of interaction effects. The correlation between the changes in dynamic steady-state balance performance (i.e. ΔTAE as TAE(T2)-TAE(T1)) and muscle activities (i.e. ΔiEMG as iEMG (T1)-iEMG(T2)) was quantified by Spearman correlation tests. By convention, a positive ΔTAE indicated an increased TAE at T2, whereas a positive ΔiEMG indicated a decreased iEMG at T2. The differences were normalized to the values at T1. The level of significance was set *a priori* to $p < 0.05$. Cohen's d and partial eta squared ($\eta^2_p$) were calculated to estimate effect sizes (small $\eta^2_p < 0.06$; medium: $0.06 < \eta^2_p < 0.14$; large: $\eta^2_p > 0.14$) [27].

## Results

The activity of MA was 7.9 ± 6.00% of MVC at T1 and 7.5 ± 5.4% of MVC at T2 for the jaw clenching condition, and for the non-clenching condition it was 0.4 ± 0.2% of MVC and 0.3 ± 0.2% of MVC at T1 and T2, respectively. This indicated that the participants performed the clenching tasks successfully.

The descriptive data of the TAE can be found in S1 Table. The TAE results at T1 are presented in Fig 2. The TAE was significantly higher in the jaw clenching condition than the non-clenching condition at T1 with high effect sizes ($p = 0.006$, d = 3.95). This showed that all participants had a better dynamic steady-state balance performance in jaw clenching condition than the non-biting condition at T1, which was in line with the hypothesis 1.

The balance and jaw clenching training effects are depicted in Fig 3. The ANOVA results revealed statistically significant effects for the factor time ($p < 0.001$, $\eta^2_p = 0.616$) and the factor clenching condition ($p = 0.008$, $\eta^2_p = 0.146$) with high effect sizes. Although there were no significant interaction effects between the factors time and group, the effect size was medium ($p = 0.174$, $\eta^2_p = 0.075$). There were no significant differences between the groups over two clenching conditions, but the effects sizes were high (OBT vs. CON: $p = 0.207$, d = 5.48; JBT vs. OBT: $p = 0.356$, d = 3.66, JBT vs. CON: $p = 0.214$, d = 5.50). These results indicated that the effects of jaw-clenching on dynamic steady-state balance performance persisted at T2, which supported the hypothesis 2.

The time normalized iEMGs are represented in Fig 4 and the descriptive data can be found in S1 Table. The ANOVA results showed that all muscle activity was significantly decreased at

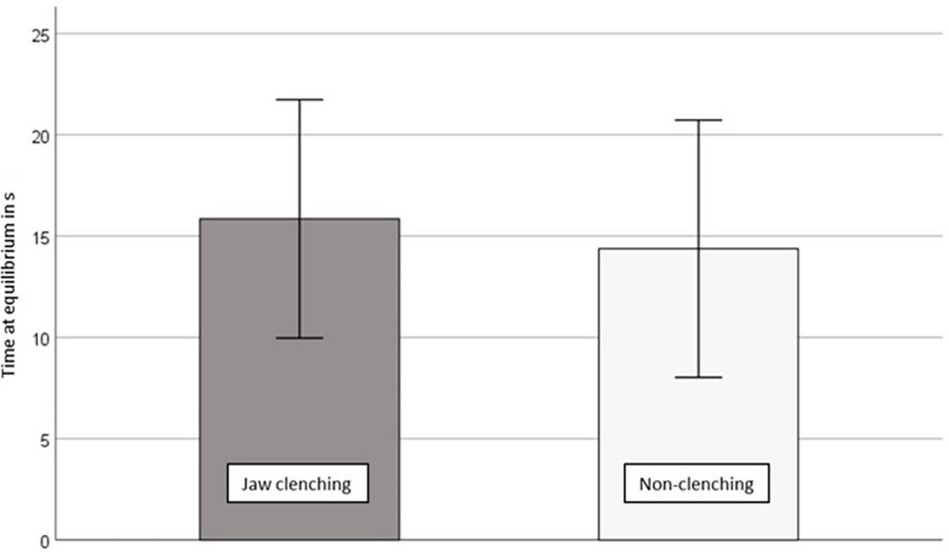

**Fig 2. Time at equilibrium for two clenching conditions at T1.**

T2 with high effect sizes (TA: $p < 0.001$, $\eta^2_p = 0.321$; GM: $p < 0.001$, $\eta^2_p = 0.289$; RF: $p < 0.001$, $\eta^2_p = 0.327$; and BF: $p < 0.001$, $\eta^2_p = 0.425$). Further, GM showed significant interaction effects between the factors time and clenching with a medium effect size (GM: $p = 0.034$, $\eta^2_p = 0.097$). These finding partly supported the hypothesis 3, since at T2 all the muscle activities decreased parallel to the dynamic steady-state balance performance improvement. However, in case of jaw clenching condition there was not any decrease in muscle activities although the dynamic steady-state balance performance was better.

The TAE increases and iEMG decreases between two measurement points are represented as the medians and 25th-75th percentiles in Table 1 [28]. The correlations between the increases in TAE and the decreases in iEMG for all muscles are also shown in Table 1. The results showed that the dynamic steady-state balance performance improvements significantly correlated with the decreases in RF activity with a moderate correlation coefficient. The rest of the muscles did not show any significant correlations.

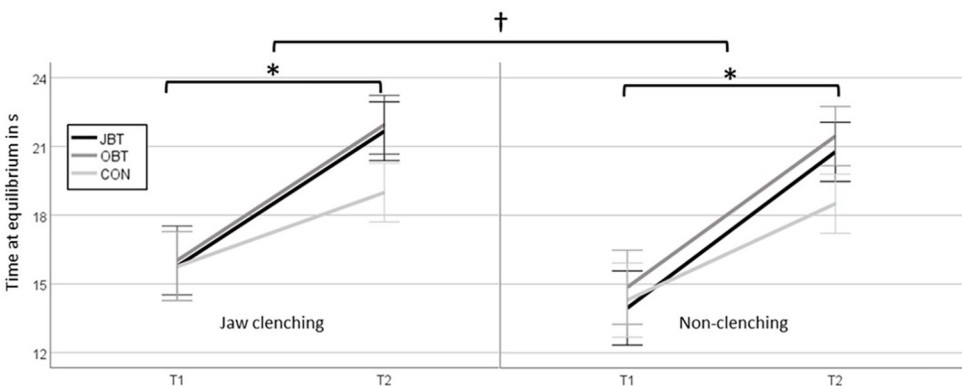

**Fig 3. Time at equilibrium for the three groups at two measurement times.** Significant differences for the factor time are indicated with * and for the factor clenching condition with †.

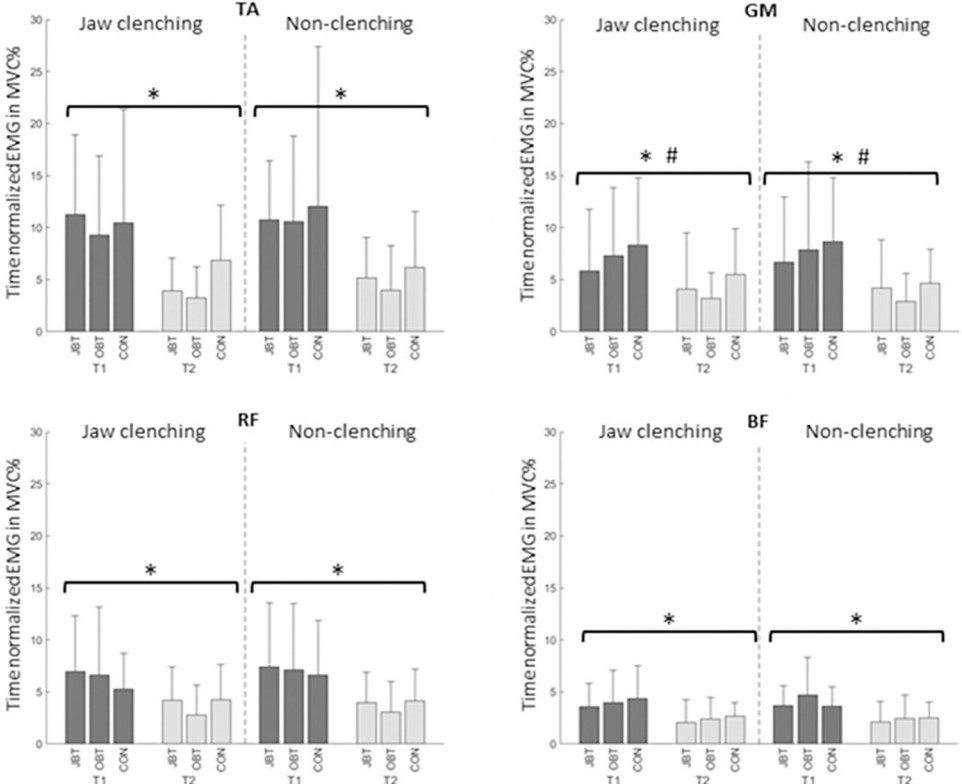

**Fig 4. Time normalized iEMGs of four muscles: Tibialis anterior (TA), gastrocnemius medialis (GM), rectus femoris (RF) and biceps femoris (BF).** JBT = jaw clenching and balance training, OBT = only balance training and CON = no-training control group. Significant differences for the factor time are indicated with * and interaction effects between the factors time and clenching condition with #.

## Discussion

This study investigated the effects of jaw clenching on dynamic steady-state balance task performance and investigated if the stabilizing effects of jaw clenching persist when the novelty of the task and the focused attention associated with it diminish. Further, activity of the selected task-relevant muscles was analyzed to better understand improvements in dynamic steady-state balance performance.

**Table 1. Time at equilibrium (TAE) increases and iEMG decreases of tibialis anterior (TA), gastrocnemius medialis (GM), rectus femoris (RF) and biceps femoris (BF) between T1 and T2, together with their correlations.**

|  | Median | 25th-75th percentile | Correlation with TAE | |
|---|---|---|---|---|
|  |  |  | p | rho |
| TAE increase in % | 36.4 | 12.1–84.1 | - | - |
| iEMG decrease in % |  |  |  |  |
| TA | 49.8 | 29.8–76.6 | 0.088 | 0.249 |
| GM | 47.0 | 23.0–65.2 | 0.054 | 0.280 |
| RF | 44.9 | 14.3–59.5 | **0.011** | **0.366** |
| BF | 41.1 | 9.1–59.0 | 0.222 | 0.179 |

The results of the clenching and non-clenching conditions were averaged for both T1 and T2.

Significant changes are shown in bold.

## Persistence of jaw clenching effects

The results showed that dynamic steady-state balance performance was better in the jaw clenching condition compared with the non-clenching condition at both T1 and T2, which was consistent with previously-shown effects during static steady-state balance [9–12]. As the effects persist at T2, it can be suggested that the performance improvements are related specifically to the jaw clenching task, but not to the novelty of the secondary task and the accompanying automatization of the balance task. Various studies have shown that jaw clenching alters postural control during upright standing [9–12], predictable perturbations, standing on the ground [14] and during unpredictable perturbations applied when standing on an oscillating platform [13]. This study complemented the previous findings by showing positive effects of jaw clenching on dynamic steady-state balance performance.

## No effects of balance training

In previous studies, training improved balance in a task-specific way [4], reduced the incidence of falls [29] and enhanced motor performance [30]. The current three-armed study design aimed to investigate the effects of simultaneous jaw clenching during balance training. Pairwise comparisons of the groups provide information on (1) if balance training alone improved the dynamic steady-state balance performance more than the no training condition (OBT vs. CON), and (2) if simultaneous jaw clenching during balance training altered the balance training effects (JBT vs. OBT) which can be explained by the automatization of the dual-task. All of the groups improved at T2 independent of their training situation. Interestingly, no significant interaction effects between the factors time and group were detected. This indicated that all groups improved their dynamic steady-state balance performance with no significant group differences. However, it should be noted that there was a medium interaction effect for TAE. Further, the *post-hoc* pairwise comparisons at T2 showed high effect sizes. The dynamic steady-state balance performance improvement, as the difference of TAE between T1 and T2 over two clenching conditions, were lower in CON by more than 2 s compared with the other two groups (JBT = 6.4 s, OBT = 6.3 s and CON = 3.7 s). Nevertheless, none of the differences reached the level of significance. Ultimately, the learning effects of the balance task were seemingly higher than the balance training effect, therefore the former outweighed the latter in terms of significance level. This finding is interesting since previous studies showed that dynamic steady-state balance performance improves after balance training comprising the same task used for testing (e.g. [5]). On the other hand, learning effects within a measurement session were also reported in previous studies in which the stabiliometer was used to quantify the dynamic steady-state balance performance [23, 31]. In this study, high learning effects of the balance task in the initial phase may have masked the effects of the balance training. For future studies, it is advisable to take more care to minimize possible learning effects when designing the study.

## Limited effects of jaw clenching on muscle activity

The iEMG results revealed that all muscle activity decreased at T2. Considering that dynamic steady-state balance performance was better at T2, it can be suggested that better performance is associated with decreased muscle activities. However, the dynamic steady-state balance performance improvements and the muscle activity reductions from T1 to T2 correlated significantly only for one of the analyzed muscles, that is RF, with a moderate correlation coefficient. The reason for the non-significant correlations may be the linear approach used both for the calculation of the changes between the two measurement sessions and for the correlations. For example, in a previous study comparing the muscle activation during back squats with

different loads showed that the correlation between the changes in the loads and the muscle activations are not linear [32]. Based on this finding, it can be suggested that the non-linear approaches for the correlation between the dynamic steady-state balance performance improvements and the iEMG reductions might reveal significant and stronger correlations. Nevertheless, all of the muscles showed reduced activities at T2 parallel to the dynamic steady-state balance performance improvement. These findings are in line with previous studies [e.g. 23] reporting practice-related reductions in muscle activations, which could relate to improved movement efficiency. On the other hand, the iEMG results in this study did not show any decrease in the jaw clenching condition, although the dynamic steady-state balance performance in the jaw clenching condition was significantly better than in the non-clenching. Further, the activity of GM decreased less at T2 in the jaw clenching condition compared with the non-clenching condition. Based on these findings, it can be suggested that dynamic steady-state balance performance improvement due to jaw clenching was not associated solely with movement efficiency, but could be explained by other mechanisms that are currently undiscovered.

## Limitations

Certain limitations of this study should be considered (1) since the participants were physically active adults, the results are not necessarily valid for other groups. (2) The best trial was taken instead of the average of five trials, since previous studies reported that the participants improved their dynamic steady- state balance performances on the stabilometer during trials on the first measurement day [23, 31]. Taking the best trials aimed to eliminate the additional effects due to different learning curves at T1 and T2. (3) Significant time effects were found even for the CON group, who did not train between two measurement times. These high learning effects may have outweighed the other effects. (4) Considering the task-specific characteristics of balance [5], it is important to add that the results are not generalizable to other static or dynamic balance tasks.

## Conclusion

This study investigated the effects of jaw clenching on dynamic steady-state balance performance across two measurement times separated by two weeks. The findings indicated that jaw clenching was associated with a better dynamic steady-state balance performance and the effects persisted even when the jaw clenching task lost its novelty and competing influence. Independent of the intervention, all groups had better dynamic steady-state balance performances at T2, which indicated high learning effects of the dynamic steady-state balance task. Moreover, learning-related reductions in muscle activity were observed at T2.

## Supporting information

**S1 Table. Time at equilibrium and time normalized iEMGs of four muscles (tibialis anterior (TA), gastrocnemius medialis (GM), rectus femoris (RF) and biceps femoris (BF)) for the three groups at two measurement times.** JBT = jaw clenching and balance training, OBT = only balance training and CON = no-training control group. The results are represented as mean ± standard deviation.
(PDF)

**S1 File. Data set.**
(XLSX)

## Author Contributions

**Conceptualization:** Steffen Ringhof, Daniel Hellmann, Thorsten Stein.

**Data curation:** Cagla Fadillioglu, Lisa Kanus, Felix Möhler.

**Formal analysis:** Cagla Fadillioglu.

**Funding acquisition:** Steffen Ringhof, Marc Schmitter, Thorsten Stein.

**Investigation:** Cagla Fadillioglu, Daniel Hellmann, Thorsten Stein.

**Methodology:** Cagla Fadillioglu.

**Project administration:** Marc Schmitter, Daniel Hellmann, Thorsten Stein.

**Resources:** Daniel Hellmann, Thorsten Stein.

**Software:** Cagla Fadillioglu.

**Supervision:** Steffen Ringhof.

**Visualization:** Cagla Fadillioglu.

**Writing – original draft:** Cagla Fadillioglu.

**Writing – review & editing:** Lisa Kanus, Felix Möhler, Steffen Ringhof, Marc Schmitter, Daniel Hellmann, Thorsten Stein.

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
