## [Decision Letter · Decision Letter 0]

14 Dec 2023

PONE-D-23-35640Persisting effects of jaw clenching on dynamic steady-state balancePLOS ONE

Dear Dr. Fadillioglu,

Thank you for submitting your manuscript to PLOS ONE. After careful consideration, we feel that it has merit but does not fully meet PLOS ONE’s publication criteria as it currently stands. Therefore, we invite you to submit a revised version of the manuscript that addresses the points raised during the review process.

**ACADEMIC EDITOR: **

This paper is interesting.

We look forward to receiving your revised manuscript.

Kind regards,

Rocco Franco

Academic Editor

PLOS ONE

“This work was supported by the German Research Foundation [STE 2093/4-3 & SCHM 2456/6-3].”

4. We note that Figure 1 in your submission contain copyrighted images. All PLOS content is published under the Creative Commons Attribution License (CC BY 4.0), which means that the manuscript, images, and Supporting Information files will be freely available online, and any third party is permitted to access, download, copy, distribute, and use these materials in any way, even commercially, with proper attribution. For more information, see our copyright guidelines: http://journals.plos.org/plosone/s/licenses-and-copyright.

Additional Editor Comments:

Dear Authors,

Please revise this paper according reviewer's recommendation

Regards

Reviewers' comments:

Reviewer's Responses to Questions

**Comments to the Author**

1. Is the manuscript technically sound, and do the data support the conclusions?

Reviewer #1: Yes

Reviewer #2: Yes

2. Has the statistical analysis been performed appropriately and rigorously? 

Reviewer #1: Yes

Reviewer #2: No

3. Have the authors made all data underlying the findings in their manuscript fully available?

Reviewer #1: Yes

Reviewer #2: No

4. Is the manuscript presented in an intelligible fashion and written in standard English?

Reviewer #1: Yes

Reviewer #2: Yes

5. Review Comments to the Author

Reviewer #1: Manuscript entitled “Persisting effects of jaw clenching on dynamic steady-state balance” aiming to evaluate jaw clenching exercises on dynamic steady state. The results showed positive effects of jaw clenching on dynamic steady-state performance. The topic is interesting and relevant in the balance training field. Overall, the manuscript is written in a good detail and quality and is acceptable after strengthening flow of idea.

Comments:

Please elaborate flow of idea more.

EMG measurements needs to be explained in more detail.

Post hoc-tests is recommended to change to t(holm).

Reviewer #2: I would like to express my gratitude to the editor and authors for providing me the opportunity to review this manuscript.

General Observation

This study presents an investigation into three research questions related to jaw clenching and dynamic steady-state balance. The utilization of Time at Equilibrium (TAE) as an indicator of dynamic-state balance, alongside measurements of lower limb EMG activities, forms a robust basis for analysis. The findings indicating no significant effects of balance training, with or without jaw clenching, on dynamic steady-state balance performance at T2 are particularly noteworthy. The observation of reduced EMG activities in all study groups at T2 is also of interest.

Specific Comments

Abstract

The term "dynamic steady-state balance performance" is not uniformly applied throughout the manuscript. In some instances, it is referred to simply as "dynamic steady-state performance." Consistency in terminology is crucial for clarity, and I recommend standardizing the use of the full term throughout the document unless differentiation is explicitly required.

Introduction

The introduction highlights the novel aspect of this study, focusing on the effects of jaw clenching on dynamic steady-state balance performance. To enhance reader comprehension, it would be beneficial to provide a clear definition of "dynamic steady-state balance performance," ideally with an illustrative example. This will help in distinguishing this study's focus from previous research.

Methods

The third objective of the study aims to investigate the association between changes in EMG activities and improvements in dynamic-state balance. However, the manuscript lacks statistical analysis directly addressing this objective. I recommend conducting and including correlational analyses to examine the relationship between EMG activity changes and TAE improvements.

Results

While the main results are summarized in the text and figures, detailed data such as specific values and standard deviations for each study group at both T1 and T2, especially for TAE and iEMG, are missing. Incorporating tables with exact values and 95% confidence intervals for group and time differences would significantly enhance the comprehensibility and transparency of the results.

Discussion

The manuscript suggests a possible group x time effect, evidenced by a medium effect size for the interaction between these factors, despite the lack of statistical significance. This claim should be approached with caution due to the potential risk of a Type I error, as the study was predicated on a sample size calculation for this specific analysis. A more informative approach might involve discussing why this effect was not statistically significant in the context of this study, perhaps by comparing the study design, outcomes, and the nature of the balance task with previous research.

6. PLOS authors have the option to publish the peer review history of their article (what does this mean?). If published, this will include your full peer review and any attached files.

Reviewer #1: **Yes: **Hamed Esmaeili

Reviewer #2: **Yes: **Yosuke Tomita

---

## [Author Response · Author response to Decision Letter 0]

10 Jan 2024

Academic editor

The authors thank the academic editor for his comment. Our responses regarding the journal requirements are summarized below:

1. Thank you for the templates. We revised our manuscript accordingly.

2. The funders had no role in study design. We added this in the manuscript.

3. We uploaded the minimal underlying data set (S1 File) and added the information, where it can be found (Under the supporting information section.)

4. We decided to replace the Figure 1 (without the lower legs, since they do not give any critical information) and uploaded in the submission system.

5. We replaced the reference [18], which was [17] in the initial version, with a newer one. The initial one was the original paper but it was from the year 1885 and was written in german.

Old: Jendrassik, E. (1885). Zur untersuchung des kniephaenomens. Neur Zbl., 4, 412–415.

New: Gregory, J. E., Wood, S. A., & Proske, U. (2001). An investigation into mechanisms of reflex reinforcement by the Jendrassik manoeuvre. Experimental Brain Research, 138(3), 366–374. https://doi.org/10.1007/s002210100707

Please let us know if further updates would be necessary. To the best of our knowledge, none of the references are retracted and the reference list is complete and correct.

Reviewer #1

The authors thank the reviewer for the critical revision of our work and for his comments. We revised our manuscript carefully to address his points. We hope we successfully comprised all the aspects. We highlighted the changes made in turquois. Please see below for our responses to each point.

Please elaborate flow of idea more.

Thank you for this critical comment. We revised the manuscript to improve the flow of the idea. More concretely, we added the short definitions of the single sub-categories of the balance (Line 55ff), to make the reader easily distinguish between the sub-categories. Later we added a sentence to emphasize the focus of the study (Line 89f). Further, following the suggestion of the other reviewer, we added the correlation analysis between the changes is in time at equilibrium and iEMG of the analyzed muscles. Ultimately, we discussed the results in more detail (Line 274ff). We hope our manuscript became better now in terms of the flow of the idea.

EMG measurements needs to be explained in more detail.

Thank you very much for this comment. We added the correlation analysis between the changes in dynamic steady-state balance performance and muscle activity which was quantified by Spearman correlation tests (Line 177ff) as suggested by the other reviewer. We summed up the results in the results section (Line 218ff) and explained the EMG measurements in discussion section in more detail (Line 274ff) particularly in combination with the time at equilibrium changes.

Post hoc-tests is recommended to change to t(holm).

Thank you very much for pointing out this point. The post-hoc results are now corrected by using the Bonferroni-Holm method. We added this information in the methods section (Line 177) and changed the p-values (Line 201f). Please note that the level of significance of the results was not affected by the correction. 

Reviewer #2

The authors thank the reviewer for the critical revision of our work and for his comments. We revised our manuscript carefully to address his points. We hope we successfully comprised all the aspects. We highlighted the changes made in turquois. Please see below for our responses to each point.

Abstract

The term "dynamic steady-state balance performance" is not uniformly applied throughout the manuscript. In some instances, it is referred to simply as "dynamic steady-state performance." Consistency in terminology is crucial for clarity, and I recommend standardizing the use of the full term throughout the document unless differentiation is explicitly required.

Thank you very much for pointing out this point. We changed all of the instances to "dynamic steady-state balance performance" to make the terms consistent throughout the manuscript.

Introduction

The introduction highlights the novel aspect of this study, focusing on the effects of jaw clenching on dynamic steady-state balance performance. To enhance reader comprehension, it would be beneficial to provide a clear definition of "dynamic steady-state balance performance," ideally with an illustrative example. This will help in distinguishing this study's focus from previous research.

Thank you for this suggestion. We added the explanation for all of the four sub-categories of balance in the first paragraph of the introduction (Line 55-60). We hope now it is easier for reader to distinguish between the sub-categories. We would like refrain from adding an illustrative example, since the stabilometer task is already shown in the Figure 2. To give more examples might be even more confusing. We hope this argumentation is plausible.

Methods

The third objective of the study aims to investigate the association between changes in EMG activities and improvements in dynamic-state balance. However, the manuscript lacks statistical analysis directly addressing this objective. I recommend conducting and including correlational analyses to examine the relationship between EMG activity changes and TAE improvements.

Thank you very much for this critical comment. We included now the the correlation analysis between the changes in dynamic steady-state balance performance and muscle activity, which was quantified by Spearman correlation tests. Please note that in this analysis by convention, a positive ∆TAE indicated an increased TAE at T2, whereas a positive ∆iEMG indicated a decreased iEMG at T2 (Line 178ff). We explained the results in the results sections (Line 218ff). In summary, the analyses show that the dynamic steady-state balance performance improvements and the muscle activity reductions from T1 to T2 correlated significantly only for one of the analyzed muscles, that is RF, with a moderate correlation coefficient. The reason for the non-significant correlations may be the linear approach used both for the calculation of the changes between the two measurement sessions and for the correlations. We discussed the results in the discussion section in more detail (Line 274ff)

Results

While the main results are summarized in the text and figures, detailed data such as specific values and standard deviations for each study group at both T1 and T2, especially for TAE and iEMG, are missing. Incorporating tables with exact values and 95% confidence intervals for group and time differences would significantly enhance the comprehensibility and transparency of the results.

Thank you very much for this suggestion. We included the descriptive data in S1 Table under the supporting information (Line 433ff). We represented only the means and standard deviations, but not the confidence intervals and the ANOVA results in the table to avoid any redundancy in information. Further, we added also the minimal data set as required by the Plos ONE journal (https://journals.plos.org/plosone/s/data-availability#loc-minimal-data-set-definition) in S1 File as excel sheets (Line 438). We hope the comprehensibility and transparency of the results became better now.

Discussion

The manuscript suggests a possible group x time effect, evidenced by a medium effect size for the interaction between these factors, despite the lack of statistical significance. This claim should be approached with caution due to the potential risk of a Type I error, as the study was predicated on a sample size calculation for this specific analysis. A more informative approach might involve discussing why this effect was not statistically significant in the context of this study, perhaps by comparing the study design, outcomes, and the nature of the balance task with previous research.

Thank you very much for this critical comment. We removed the sentence indicating that a larger sample size could be beneficial. We added the sentence to emphasize that the high learning effects were the biggest issue and suggested that for future studies, it is advisable to take more care to minimize possible learning effects when designing the study (Line 260ff).

---

## [Decision Letter · Decision Letter 1]

5 Feb 2024

Persisting effects of jaw clenching on dynamic steady-state balance

PONE-D-23-35640R1

Dear Dr. Fadillioglu,

We’re pleased to inform you that your manuscript has been judged scientifically suitable for publication and will be formally accepted for publication once it meets all outstanding technical requirements.

Kind regards,

Rocco Franco

Academic Editor

PLOS ONE

Additional Editor Comments (optional):

This paper is revised according to Reviewer's recommendation. This paper is ready for publication in PlosOne.

Regards

Reviewers' comments:

Reviewer's Responses to Questions

**Comments to the Author**

1. If the authors have adequately addressed your comments raised in a previous round of review and you feel that this manuscript is now acceptable for publication, you may indicate that here to bypass the “Comments to the Author” section, enter your conflict of interest statement in the “Confidential to Editor” section, and submit your "Accept" recommendation.

Reviewer #1: All comments have been addressed

Reviewer #2: All comments have been addressed

2. Is the manuscript technically sound, and do the data support the conclusions?

Reviewer #1: Yes

Reviewer #2: Yes

3. Has the statistical analysis been performed appropriately and rigorously? 

Reviewer #1: Yes

Reviewer #2: Yes

4. Have the authors made all data underlying the findings in their manuscript fully available?

Reviewer #1: Yes

Reviewer #2: Yes

5. Is the manuscript presented in an intelligible fashion and written in standard English?

Reviewer #1: Yes

Reviewer #2: Yes

6. Review Comments to the Author

Reviewer #1: Thanks for your amendment. the manuscript has been improved. I have no further comments, Congratulations to your study.

Reviewer #2: I appreciate the chance to review this engaging manuscript. The authors have successfully addressed all of my concerns, leading me to recommend that the paper be accepted.

7. PLOS authors have the option to publish the peer review history of their article (what does this mean?). If published, this will include your full peer review and any attached files.

Reviewer #1: **Yes: **Hamed Esmaeili

Reviewer #2: **Yes: **Yosuke Tomita

---

## [Editor Report · Acceptance letter]

13 Feb 2024

PONE-D-23-35640R1 

PLOS ONE

Dear Dr. Fadillioglu, 

I'm pleased to inform you that your manuscript has been deemed suitable for publication in PLOS ONE. Congratulations! Your manuscript is now being handed over to our production team.

Kind regards, 

on behalf of

Dr. Rocco Franco 

Academic Editor

PLOS ONE